# Pregnancy complications and cardiovascular disease risk perception: A qualitative study

**Prabha H. Andraweera**[1,2,3]☯*, **Zohra S. Lassi**[1,2]☯, **Maleesa M. Pathirana**[1,2], **Michelle D. Plummer**[1,2], **Gus A. Dekker**[1,2,4], **Claire T. Roberts**[1,2,5], **Margaret A. Arstall**[1,3]

**1** Adelaide Medical School, The University of Adelaide, Adelaide, Australia, **2** Robinson Research Institute, The University of Adelaide, Adelaide, Australia, **3** Department of Cardiology, Lyell McEwin Hospital, Elizabeth Vale, Australia, **4** Division of Women's Health, Lyell McEwin Hospital, Elizabeth Vale, Australia, **5** Flinders Health and Medical Research Institute, Flinders University, Bedford Park, South Australia, Australia

☯ These authors contributed equally to this work.
* prabha.andraweera@adelaide.edu.au

**Data Availability Statement:** Ethics approval was obtained from the Central Adelaide Local Health Network Human Research Ethics Committee (HREC/14/TQEH/277). The authors did not seek

## Abstract

### Objectives

We aimed to assess women's perceptions on the long-term risks for cardiovascular disease (CVD) after major pregnancy complications.

### Methods

Women who experienced major pregnancy complications and those who experienced uncomplicated pregnancies were invited to participate in a qualitative study. Focus group discussions (FGDs) and self-administered questionnaires were used to explore: The knowledge of long-term sequelae after experiencing a major pregnancy complication; Importance of education on heart health; The practicality of referral to a clinic after pregnancy complications; Willingness for regular postpartum clinic visits after pregnancy complications. A thematic qualitative analysis was undertaken.

### Results

26 women participated in four FGDs. The majority of women did not know of the association between major pregnancy complications and CVD. The main views expressed were: Women who experience pregnancy complications should receive education on improving heart health; An appointment for the first CVD risk screening visit needs to be made prior to discharge from the delivery suite; Women will benefit by having the option to select between a hospital and a general-practitioner based model of follow up.

### Conclusions

These views are important in developing postpartum strategies to reduce CVD risk among women who experience pregnancy complications.

specific consent from participants for sharing their data publicly. However, the authors invite applications to use the collected data via email to the University of Adelaide Research Branch at raohealth@adelaide.edu.au. Applicants will be asked to complete a Research Application Form specifying details for their planned study which will then be reviewed by the team at the University of Adelaide Research Office. The authors are keen to promote collaboration among researchers and to see their data used in studies which meet their ethics and consenting process.

**Funding:** NHMRC Peter Doherty Early Career Fellowship awarded to PHA (GNT1090778)., NHMRC Public Health Early Career Fellowship awarded to ZSL (GNT1141382), Robinson Research Institute Innovative Seed Funding project grant awarded to PHA, MAA, ZSL and CTR. Australian Health Research Alliance, Women's Health Research Translation Network EMCR Award (PHA). CTR is supported by a NHMRC Investigator Grant (GNT1174971) and a Matthew Flinders Fellowship from Flinders University. The funders had no role in study design, data collection and analysis, decision to publish, or preparation of the manuscript.

**Competing interests:** No authors have competing interests.

# Introduction

The link between pregnancy complications and later life cardiovascular disease (CVD) is now well established. Women who develop major pregnancy complications including gestational hypertension, preeclampsia, gestational diabetes mellitus (GDM), those who deliver small for gestational age (SGA) infants or deliver preterm are at approximately 2–3 times the risk of developing CVD compared to women who have uncomplicated pregnancies [1]. Recent systematic reviews and meta-analyses also demonstrate that risk factors for CVD are increased among children exposed to these pregnancy complications *in utero* compared to children born to uncomplicated pregnancies [2, 3]. Programming in response to an adverse intrauterine environment is one plausible mechanism that links pregnancy complications and CVD risk among children [4]. Shared genetics, lifestyle and behavioural factors are two other possible mechanisms that explain the association between pregnancy complications and CVD among women as well as their children. Genetic polymorphisms that are risk factors for CVD are more prevalent among women who experience preeclampsia, and those who give birth to SGA infants or deliver preterm; among men who father preeclamptic or SGA pregnancies and among children born to preeclamptic pregnancies or born SGA providing evidence for a genetic predisposition to pregnancy complications and CVD [5, 6]. The risk attributed to genetics is not modifiable, hence, emphasis must be on modifiable risks.

The evidence on the link between pregnancy complications and CVD resulted in the inclusion of "hypertensive diseases of pregnancy" as a major CVD risk factor for women in the 2011 American Heart Association guidelines for the prevention of heart disease in women [7, 8]. However, National guidelines in most countries do not include routine postpartum follow up and CVD risk screening for women who experience major pregnancy complications and most countries including Australia do not have a funded model of postpartum care. This is partly due to the paucity of postpartum intervention trials to guide evidence-based CVD risk reduction strategies. The limited evidence mainly focuses on preeclampsia, comes from research-based interventions and the degree of risk reduction is uncertain [9]. A hospital-based intervention offered to women at risk, as standard care is likely to be more effective and provide a pathway for long term monitoring and management of those at high risk. In order to develop sustainable and feasible CVD risk reduction strategies, views of health consumers are vital. We conducted a qualitative study to assess knowledge about the risk for CVD and willingness to attend postpartum follow up clinics after complicated pregnancies among a group of women from the northern suburbs of Adelaide.

# Material and methods

A qualitative descriptive study was undertaken to obtain a detailed source of explanatory data regarding CVD risk perception and willingness to attend postpartum follow up. The qualitative descriptive method is a distinctive and valuable technique that offers a direct description of phenomena that are still interpretive and it addresses both descriptive and interpretive validity to account for participants' observations and their understandings of these events [10]. This study was undertaken at the Lyell McEwin Hopsital, a tertiary care facility located in the northern metropolitan area of Adelaide, South Australia, which serves a community with a very low socioeconomic status and a high burden of disease [11]. The women who participated in this study had attended antenatal care at the same hospital and had been recruited to a pregnancy cohort study during their first pregnancies. They had delivered their babies at the same hospital and were invited to take part in this study while taking part in a 10-year postpartum follow up study. Women who experienced pregnancy complications, as well as those who had uncomplicated pregnancies, were invited to participate in this qualitative study. Ethics

approval was obtained from the Central Adelaide Local Health Network Human Research Ethics Committee (HREC/14/TQEH/277) and all participants provided written informed consent.

Participants were given an option of choosing a convenient time based on the availability of the venue, and a mutually convenient time was arranged to conduct the focus group discussions (FGDs). A total of 26 women attended four in-depth, semi structured, face-to-face FGDs with 6–7 women in each group. The interview guide was developed by the study team (S1 File), all discussions were facilitated by a team member, and the key areas explored included:

- The knowledge of long-term sequelae after experiencing a major pregnancy complication

- Importance of education on heart health

- The practicality of referral to a clinic after pregnancy complications

- Willingness for regular postpartum clinic visits after pregnancy complications

All interviews were audio-recorded and transcribed verbatim. The interview transcripts were then exported to NVIVO 10 [12], the qualitative software used for coding and analysis. Coding helped to organize the data, and in creating the linkages within and between the perceptions and experiences presented in the data. The data coding was continued until the theoretical saturation was reached and no new concepts emerged from successive reviewing and coding. The codes were then considered complete and emerging themes were then developed [13]. Saturation was reached after four FGDs with a total of 26 participants. Where appropriate, we used verbatim quotations from interview transcripts to illustrate responses related to relevant themes. We used a deductive approach for content analysis of our FGDs. The process involved familiarization with the material, identifying a thematic framework, indexing, charting, mapping and interpretation [14].

## Results

Twenty six women participated in four FGDs. Of these women, 13 (50%) had experienced a major pregnancy complication (preeclampsia, SGA pregnancy, spontaneous preterm birth or GDM) during the first pregnancy and 13 (50%) had experienced an uncomplicated first pregnancy. The participants were aged between 29.7 to 50.9 years (mean 38.9 years). The majority of women were married or in a stable relationship (n = 23, 88.5%). Five participants (19.2%) had completed a university degree, 15 (57.7%) had completed year 12, and the remaining six (23.1%) had completed year 10/11. The majority of participants (16/26, 61.5%) were employed. Ten (38.5%) participants had an annual family income of >$70,000 and six (23.1%) had an annual family income between $40,000–70,000 which is lower than the average annual household annual income in South Australia [15].

### Link between pregnancy complications and heart disease

Overall women were aware that pregnancy complications had adverse impacts on health but did not have any knowledge about specific adverse impacts. One participant commented that she knew the complications,

"*could lead further down the track, but not in great detail.*"

Another participant commented that,

"*I knew it would affect the child more than it would affect me....*"

The women also thought about the immediate risks during pregnancy but not those later in life. The women who experienced GDM were aware to some extent about their risk of type 2 diabetes in the future, based on the following testimonies by two participants,

> "*Maybe I have a tendency of getting type 2 (diabetes). So far no, but maybe.*"

> "*I had diabetes during my second pregnancy and I got a heap of information about that and that I have to go for a test every two years.*"

Of the 26 participants, 08 (31%) were not aware of the common risk factors for heart disease and, more than half of the participants (16/26, 62%) did not know about the link between pregnancy complications and heart disease. A participant commented,

> "*I didn't even think there was a link. . ..I just thought family history (is associated). We've got a family history but I didn't think pregnancy would cause it.*"

## Educating women about heart health after a complicated pregnancy

Unless women experienced pregnancy complications, it seemed that they were not interested in knowing more about the risks after pregnancy complications, as highlighted in the following testimonies by two participants, one of whom had an uncomplicated pregnancy and the other experienced a complicated pregnancy,

> "*Most of my first pregnancy was relatively straightforward. . .if I had an issue, I felt the staff here would find me the information, but I didn't necessarily go in search of other information because I didn't have those conditions.*"

> "*As soon as someone says you have high blood pressure, you start listening.*"

When asked about the information provided during pregnancy, some women responded that they felt overwhelmed with the information given to them. One participant commented,

> "*There was so much information given to you during pregnancy and I just thought, I don't remember, it's probably in those pamphlets. There was so much information, it's like information overload.*"

Information on risk for heart disease was something that women requested as part of routine advice during the antenatal period. Nearly all participants (25/26, 96.1%) agreed that pregnancy was a good period to receive advice on improving heart health. One woman stated,

> "*I think it's very important to know.*"

In general, women felt that if this information was provided to them, it would give them a sense of control over their pregnancy and health. One woman commented,

> "*Give me all the information, let me know what I need to do to prevent that (heart problems)*",

and another followed on by stating,

> "*Then (after receiving information) you can prevent it, you can see your GP regularly.*"

When asked if women actually received any information or advice from a doctor on how to improve heart health, the majority of participants reported they did not (17/26, 65%).

Two participants commented on how to provide this information to young mothers,

*"I think for first time mums, young mums, paper free communication is the direction. I know I'd like to access information that way."*

*"Brochures are a waste of time; you get so many of them . . .. I think I listened a lot more to the midwife when I had my appointments . . .."*

## The practicality of referral to a clinic after pregnancy complications

Many of the participants responded positively to the concept of postpartum referral to a heart health clinic for women who experience pregnancy complications. When asked if they would be happy to attend a regular clinic if referred by their obstetrician, the majority (24/26, 92%) of the participants responded 'yes'. Women's responses to "referral to a hospital-based clinic" vs "general practitioner care" varied. Some women felt that a hospital-based clinic would be more effective, as women were used to the hospital-based model following antenatal care. They felt a stronger sense of rapport and trust towards a hospital-based clinic. This was mentioned in a comment by one participant,

*"If it's something the hospital has detected, I'd rather come back to the hospital's women's health unit."*

Some of the participants who preferred the idea of a hospital-based clinic mentioned that they felt instability in a GP based model as they could not always find a GP with whom they were comfortable and that the GP might not be aware of details about their pregnancy. One woman stated that,

*"I would rather have the same doctor, so they know your history, you're not seeing a new doctor each time, you've got to refill them information of what is going on."*

However, some women were supportive of the idea of a GP-based model, stating that there would be more flexibility in appointment times and it was more suitable for busy lifestyles. A participant commented on why a GP may be more preferable,

*"I think a GP would be probably more convenient in many ways, there is quite often more options on times where you can go."*

One participant commented on the advantages of either a GP-based or hospital-based follow up,

*"Thinking sort of a bit on both, yes GP on access, availability, you can sort of suit around your (life) structure a bit more, but if its (hospital) clinic it's more focused. So I think a clinic would be better for that but a GP better for fitting it into your life (and routine)."*

Some were of the opinion that having the option to select between the two would be the ideal, as commented by one participant:

*"Being given an option between clinic (in hospital) and GP is probably the ideal situation."*

## Willingness for regular postpartum clinic visits after pregnancy complications

When discussing whether young women would be happy to come in for regular visits to a heart health clinic after pregnancy complications, many participants thought that heart health should be integrated into a model of regular postpartum health care. They also suggested that scheduled appointments would be more effective than referrals. One participant commented on methods of improving clinic attendance,

"*Don't ask them if they want to; tell them this is your appointment. And you persuade them that way and you get them to come rather than giving them the choice.*"

Another participant commented,

"*Make an appointment, make it here in women's health and then . . . remind us, like a text message or a phone call or a letter.*"

The participants felt that if these appointments were scheduled by the hospital, young women would be more obliged to listen and take their health seriously. One participant stated:

"*I think it's more for them to treat me, being responsible, ok I need to take my medication. . .*"

Two participants commented on the importance of engaging with women during pregnancy,

"*If you train the mothers from their first pregnancy, where they are used to being told what to do by the hospital, they'll come back because it's part of the process.*"

"*If it's the hospital telling you what to do, well then I'd better come back.*"

Women felt that time of the follow up clinic visit was an important consideration during the postpartum period. One participant suggested having the first heart health clinic appointment at the same time as the baby's 6-week check-up visit,

"*You know the six week check-up that the baby has, maybe then, you come in for your baby, you get checked, go over everything, doing it at the same time I think that would help.*"

Some participants felt that these clinics could be offered in a similar manner to antenatal classes in groups. A participant commented on the benefit of allocating these clinics by age,

"*Maybe they [women] would feel more comfortable if the antenatal group was all their own age, or similar, whereas if they were seeing an older mum they sort of think, oh, she'd probably think I'm going to be stupid if I ask that question or they might be even thinking that I'm too young to be having a baby. But if they're all the same age group then they're going to know that everyone's in the same boat.*"

One participant commented that:

"*Pre-natal care can follow into the post-natal information clinics.*"

The women felt that having support from other young mothers will allow women to gain a sense of support and become more responsible for their health.

"*When you've got people and you form friendships they're going to help you stay accountable for your health, for follow up appointments, they're going to say, hey are you coming to this appointment? Come, you've got to look after yourself for bub (baby). So you've got someone encouraging, you've got your group encouragement whereas some of these girls (young mothers) might not otherwise have that support network.*"

## Discussion

This qualitative study assessed the knowledge of young women on the association between major pregnancy complications and later life CVD risks and sought their views on strategies to improve cardiovascular health during the postpartum period for women who experience major pregnancy complications.

Most women did not know of the increased risk for CVD after experiencing major pregnancy complications and 31% were not aware of the common risk factors for CVD. Women who experienced GDM were aware of the increased risk of type 2 diabetes after experiencing GDM and were aware of the importance of regular screening after pregnancy. However, those who experienced other pregnancy complications including preeclampsia were not aware of any health risks beyond pregnancy. Women who experience major pregnancy complications including preeclampsia, gestational hypertension, GDM, as well as women who give birth to SGA infants and those who experience spontaneous preterm labour are at approximately twice the risk of CVD compared to women who experience uncomplicated pregnancies [1]. Therefore, all women who experience these pregnancy complications should receive education on the risk and have the opportunity for early screening to detect any modifiable risk factors before CVD manifests.

The women felt that providing health education on improving cardiovascular health after pregnancy complications to all women during the antenatal period is not useful as only those who experience pregnancy complications will be interested in such information. They also thought that paper-based educational material such as brochures will not be read by many women. Moreover, providing information on heart health to women during the antenatal period will not be considered a feasible option by health care providers as the focus of care during this time is achieving the best outcome in pregnancy. The women suggested that if group activities similar to antenatal classes could be organised during the postpartum period, they will be of interest to many young women and will be beneficial in engaging women for CVD risk screening.

The women suggested that an appointment for the first CVD risk screening should be made prior to discharge from the delivery suite. They also thought that if the first CVD risk screening visit could be arranged to coincide with the 6 week health check of the baby, most young women would comply, and that it will help improve attendance rates at subsequent follow up visits. If the 6 week health checks of the baby are not conducted in a hospital setting, the CVD screening visit could be organised at a facility that performs the health checks of the baby. In addition, a telehealth appointment may also provide an alternative approach in order to ensure that regular follow up in maintained.

Mixed views were discussed in response to acceptability and feasibility of a hospital-based postpartum screening model compared to a general-practitioner based model. Being familiar with the hospital immediately after the antenatal period, and the hospital having records of details of the pregnancy were all considered as advantages of a hospital-based model, which

would increase clinic attendance, whereas flexibility with appointments was considered an advantage of the GP-based model. The women thought that, if possible, having the option of selecting either of the two methods was the best. Consistent with the views expressed by the women, a previous randomised controlled trial that evaluated the proportion of uptake of GDM screening in the primary vs secondary care setting showed that 14.8% of those randomised to primary care did not attend screening compared to 8.5% of those randomised to secondary care (95% CI 1.9% to 10.8%, p 0.005) [16].

Overall, the views expressed by women are important in planning postpartum CVD risk reduction strategies for those who experience major pregnancy complications. The limitations in our study are: the women were recruited from those taking part in a 10-year postpartum follow-up study, hence the views may be subject to selection bias. The views of women 10 years postpartum may not represent the views of younger women, especially those in the early postpartum period, however, most women who participated in the interviews had had a subsequent pregnancy 3–5 years prior to the interviews. In addition, the participants were recruited from a hospital which serves a community with a very low socioeconomic status, hence the views of the participants will not represent the views of the majority of women in Australia. However, the strengths of the study were that the women who took part in the interview had attended the same hospital for antenatal care and hence their views on continuous care were very important. Also, 50% the participants had experienced a major pregnancy complication and the others had experienced uncomplicated pregnancies, thereby providing views of both consumer groups.

Replicating the FGDs amongst a group of women within the first postpartum year will provide the opportunity to obtain views from a younger cohort. In addition, including women who seek health care in both public and private sector hospitals as well as conducting the FGDs in hospitals in different locations will provide information from participants of different socioeconomic backgrounds. These consumer views will be vital in planning postpartum CVD risk reduction strategies that will be feasible and acceptable to young women who experience major pregnancy complications.

## Supporting information

**S1 File. Interview guide.**
(DOCX)

## Acknowledgments

The authors wish to thank all the study participants.

## Author Contributions

**Conceptualization:** Prabha H. Andraweera, Zohra S. Lassi, Gus A. Dekker, Claire T. Roberts, Margaret A. Arstall.

**Formal analysis:** Prabha H. Andraweera.

**Funding acquisition:** Prabha H. Andraweera.

**Investigation:** Michelle D. Plummer.

**Methodology:** Prabha H. Andraweera, Zohra S. Lassi.

**Project administration:** Prabha H. Andraweera.

**Resources:** Maleesa M. Pathirana.

**Software:** Zohra S. Lassi.

**Writing – original draft:** Prabha H. Andraweera, Maleesa M. Pathirana.

**Writing – review & editing:** Prabha H. Andraweera, Zohra S. Lassi, Maleesa M. Pathirana, Michelle D. Plummer, Gus A. Dekker, Claire T. Roberts, Margaret A. Arstall.

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
