## [Decision Letter · Decision Letter 0]

31 May 2022

PONE-D-21-11952Pregnancy complications and cardiovascular disease risk perception: A qualitative studyPLOS ONE

Dear Dr. Andraweera,

Thank you for submitting your manuscript to PLOS ONE. After careful consideration, we feel that it has merit but does not fully meet PLOS ONE’s publication criteria as it currently stands. Therefore, we invite you to submit a revised version of the manuscript that addresses the points raised during the review process.

Please note that we have only been able to secure a single reviewer to assess your manuscript. We are issuing a decision on your manuscript at this point to prevent further delays in the evaluation of your manuscript. Please be aware that the editor who handles your revised manuscript might find it necessary to invite additional reviewers to assess this work once the revised manuscript is submitted. However, we will aim to proceed on the basis of this single review if possible.

We look forward to receiving your revised manuscript.

Kind regards,

Hugh Cowley

Senior Editor

PLOS ONE

Journal Requirements:

2. Please include a copy of the interview guide used in the study, in both the original language and English, as Supporting Information, or include a citation if it has been published previously.

3. Thank you for stating the following financial disclosure: "CTR is supported by a National Health and Medical Research Council Investigator Grant (GNT1174971) "

4. Thank you for stating the following in the Acknowledgments Section of your manuscript: "NHMRC Peter Doherty Early Career Fellowship awarded to PHA, NHMRC Public Health Early Career Fellowship awarded to ZSL, Robinson Research Institute Innovative Seed Funding project grant awarded to PHA, MAA, ZSL and CTR. CTR is supported by a NHMRC Investigator Grant (GNT1174971) and a Matthew Flinders Fellowship from Flinders University."

Please remove any funding-related text from the manuscript and let us know how you would like to update your Funding Statement. Currently, your Funding Statement reads as follows: "CTR is supported by a National Health and Medical Research Council Investigator Grant (GNT1174971) "

6. Your abstract cannot contain citations. Please only include citations in the body text of the manuscript, and ensure that they remain in ascending numerical order on first mention.

Reviewers' comments:

Reviewer's Responses to Questions

**Comments to the Author**

1. Is the manuscript technically sound, and do the data support the conclusions?

Reviewer #1: Yes

2. Has the statistical analysis been performed appropriately and rigorously? 

Reviewer #1: N/A

3. Have the authors made all data underlying the findings in their manuscript fully available?

Reviewer #1: Yes

4. Is the manuscript presented in an intelligible fashion and written in standard English?

Reviewer #1: Yes

5. Review Comments to the Author

Reviewer #1: A qualitative study of women's understanding of cardiovascular risk following obstetric complications.

I gather that this is a long term study and these women were being interviewed long after their pregnancy. If that is correct, it may have some implications for their answers, can you comment on this?

I note the main finding that women would like to be seen at their hospital for their 6 weeks check as well as a counselling appointment however hospitals in my state do not conduct 6 week checks. What implications does this have for your findings? Perhaps that the information should be provided with the 6 week check wherever it may be?

I note that only face to face options have been considered in this work. DO the authors have any suggestions for virtual appointments? Was this work conducted prior to the pandemic? Might options have changed?

6. PLOS authors have the option to publish the peer review history of their article (what does this mean?). If published, this will include your full peer review and any attached files.

Reviewer #1: **Yes: **Penelope Sheehan

---

## [Author Response · Author response to Decision Letter 0]

19 Jun 2022

20th June 2022

PONE-D-21-11952

Pregnancy complications and cardiovascular disease risk perception: A qualitative study

We would like to thank reviewers for their time and effort in appraising our manuscript critically and providing valuable comments for improvements. Please find below our responses to the reviewers comments and a revised paper taking these comments into consideration.

Reviewer #1: A qualitative study of women's understanding of cardiovascular risk following obstetric complications.

I gather that this is a long term study and these women were being interviewed long after their pregnancy. If that is correct, it may have some implications for their answers, can you comment on this?

We agree with the reviewer and have included the following paragraph in the discussion.

“The limitations in our study are: the women were recruited from those taking part in a 10-year postpartum follow-up study, hence the views may be subject to selection bias. The views of women 10 years postpartum may not represent the views of younger women, especially those in the early postpartum period, however, most women who participated in the interviews had had a subsequent pregnancy 3-5 years prior to the interviews” (page 14, lines 311-316 of the revised manuscript).

I note the main finding that women would like to be seen at their hospital for their 6 weeks check as well as a counselling appointment however hospitals in my state do not conduct 6 week checks. What implications does this have for your findings? Perhaps that the information should be provided with the 6 week check wherever it may be?

We agree with the reviewer and have added the sentence below to the discussion.

“If the 6 week health checks of the baby are not conducted in a hospital setting, the CVD screening visit could be organised at a facility that performs the health checks of the baby”. In addition, a telehealth appointment may also provide an alternative approach in order to ensure that regular follow up is maintained”. (page13, lines 295-298 of the revised manuscript).

I note that only face to face options have been considered in this work. DO the authors have any suggestions for virtual appointments? Was this work conducted prior to the pandemic? Might options have changed?

We agree with the reviewer. Please response to above comment.

Additional Editor comment

Please include a copy of the interview guide used in the study, in both the original language and English, as Supporting Information, or include a citation if it has been published previously

We have now included the interviewer guide as supplementary material

Please see below the funding statement

NHMRC Peter Doherty Early Career Fellowship awarded to PHA (GNT1090778)., NHMRC Public Health Early Career Fellowship awarded to ZSL (GNT1141382), Robinson Research Institute Innovative Seed Funding project grant awarded to PHA, MAA, ZSL and CTR. CTR is supported by a NHMRC Investigator Grant (GNT1174971) and a Matthew Flinders Fellowship from Flinders University. The funders had no role in study design, data collection and analysis, decision to publish, or preparation of the manuscript.

---

## [Editor Report · Decision Letter 1]

7 Jul 2022

Pregnancy complications and cardiovascular disease risk perception: A qualitative study

PONE-D-21-11952R1

Dear Dr. Andraweera,

We’re pleased to inform you that your manuscript has been judged scientifically suitable for publication and will be formally accepted for publication once it meets all outstanding technical requirements.

Kind regards,

Gaetano Santulli, MD

Academic Editor

PLOS ONE

---

## [Editor Report · Acceptance letter]

11 Jul 2022

PONE-D-21-11952R1 

Pregnancy complications and cardiovascular disease risk perception: A qualitative study 

Dear Dr. Andraweera:

I'm pleased to inform you that your manuscript has been deemed suitable for publication in PLOS ONE. Congratulations! Your manuscript is now with our production department. 

Kind regards, 

on behalf of

Professor Gaetano Santulli 

Academic Editor

PLOS ONE